# Intracranial Gene Delivery Mediated by Albumin-Based Nanobubbles and Low-Frequency Ultrasound

**DOI:** 10.3390/nano14030285

**Published:** 2024-01-30

**Authors:** Takayuki Koga, Hiroshi Kida, Yutaro Yamasaki, Loreto B. Feril, Hitomi Endo, Keiji Itaka, Hiroshi Abe, Katsuro Tachibana

**Affiliations:** 1Department of Neurosurgery, Faculty of Medicine, Fukuoka University, 7-45-1 Nanakuma, Jonan-ku, Fukuoka 814-0180, Japan; md210012@cis.fukuoka-u.ac.jp (T.K.); hiroabe@fukuoka-u.ac.jp (H.A.); 2Department of Anatomy, Faculty of Medicine, Fukuoka University, 7-45-1 Nanakuma, Jonan-ku, Fukuoka 814-0180, Japan; kida_hiroshi@fukuoka-u.ac.jp (H.K.); yuyamasaki@adm.fukuoka-u.ac.jp (Y.Y.); feril@fukuoka-u.ac.jp (L.B.F.J.); endo63@fukuoka-u.ac.jp (H.E.); 3Department of Biofunction Research, Institute of Biomaterials and Bioengineering, Tokyo Medical and Dental University (TMDU), 2-3-10 Kanda-Surugadai, Tokyo 101-0062, Japan; itaka.bif@tmd.ac.jp

**Keywords:** nanobubble, low-frequency ultrasound, drug delivery system, central nervous system

## Abstract

Research in the field of high-intensity focused ultrasound (HIFU) for intracranial gene therapy has greatly progressed over the years. However, limitations of conventional HIFU still remain. That is, genes are required to cross the blood-brain barrier (BBB) in order to reach the neurological disordered lesion. In this study, we introduce a novel direct intracranial gene delivery method, bypassing the BBB using human serum albumin-based nanobubbles (NBs) injected through a less invasive intrathecal route via lumbar puncture, followed by intracranial irradiation with low-frequency ultrasound (LoFreqUS). Focusing on both plasmid DNA (pDNA) and messenger RNA (mRNA), our approach utilizes LoFreqUS for deeper tissue acoustic penetration and enhancing gene transfer efficiency. This drug delivery method could be dubbed as the “Spinal Back-Door Approach”, an alternative to the “front door” BBB opening method. Experiments showed that NBs effectively responded to LoFreqUS, significantly improving gene transfer in vitro using U-87 MG cell lines. In vivo experiments in mice demonstrated significantly increased gene expression with pDNA; however, we were unable to obtain conclusive results using mRNA. This novel technique, combining albumin-based NBs and LoFreqUS offers a promising, efficient, targeted, and non-invasive solution for central nervous system gene therapy, potentially transforming the treatment landscape for neurological disorders.

## 1. Introduction

High-intensity focused ultrasound (HIFU) for drug delivery to the brain is a promising area of research that involves the use of focused ultrasound to temporarily disrupt the blood-brain barrier (BBB). HIFU is non-invasive compared to conventional surgical methods and does not require incisions or the insertion of surgical instruments into the brain, therefore reducing the risk of complications and infections. Furthermore, by precisely focusing the ultrasound on a specific part of the brain, HIFU can temporarily open the BBB at desired locations, enabling drugs to reach the target more effectively [1]. Recent advancements in ultrasound-based gene delivery therapy to the brain have shown promise in enhancing the targeted and non-invasive delivery of genetic material [2,3]. Some notable developments include the use of HIFU in conjunction with microbubbles to transiently open the blood-brain barrier. This method allows for a more efficient delivery of therapeutic genes or gene editing tools into the brain. Studies have demonstrated the potential of ultrasound-assisted gene delivery in the treatment of Parkinson’s disease, Alzheimer’s disease, glioblastoma, and other neurological disorders [4]. However, there are technical challenges, such as the limited location in the brain where precise and effective drug delivery can be achieved. Furthermore, the limited amount of drug crossing the BBB can hinder effective treatments for various neurological disorders, as well as potential tissue damage or adverse effects caused by high-intensity ultrasound.

Low-frequency ultrasound (LoFreqUS) can penetrate deeper into the tissues than high-frequency ultrasonography [5]. In the context of brain therapy, deeper penetration allows for more profound effects on the targeted brain regions without significant attenuation. We have previously reported the use of 1 MHz ultrasound as a potential tool for cancer gene therapy [6,7]. Several previous studies have reported that LoFreqUS collapses microbubbles and achieves gene or drug delivery to living tissue [8,9]. Nanobubbles (NBs), officially called ultrafine bubbles [10], are bubbles with submicron diameters smaller than microbubbles. Our previous study revealed certain characteristics of human serum albumin-based NBs. High-speed shaking of a solution containing a low concentration of HSA generated a high concentration of NBs [11]. Analysis with an electron microscope of the lyophilized material demonstrated that the NBs generated by this method were encapsulated by a shell of HSA [7]. While high-frequency ultrasound is often utilized for imaging and therapeutic applications, low-frequency ultrasound in the range of 40 kHz may prove to be more efficient in conjunction with NBs size in the submicron range. We hypothesized that LoFreqUS can non-invasively penetrate deeper into tissues and perhaps propagate through the vertebrae/skull more easily, bursting more NBs, and thus resulting in efficient gene transfer into brain tissue.

The choice between plasmid DNA (pDNA) and messenger RNA (mRNA) for brain gene therapy depends on several factors, including the specific therapeutic goal, duration of gene expression needed, and the unique properties of each molecule [12]. The pDNA can provide sustained gene expression as it integrates into the host cell genome. Moreover, pDNA can continuously produce encoded proteins over an extended period. In contrast, mRNA-based therapies offer transient gene expression without genomic integration [13]. This transient expression can be advantageous for controlling the duration of protein production and minimizing long-term effects or potential immune responses. Our second hypothesis was that in the presence of NBs, LoFreqUS can enhance the transfer of both pDNA and mRNA to intracranial tissues.

Our final hypothesis concerns a novel method of naked gene delivery via the spine. Intrathecal drug delivery involves the direct administration of drugs into the intrathecal space around the spinal cord, which is filled with cerebrospinal fluid (CSF). This method bypasses the bloodstream and allows drugs to directly reach the central nervous system (CNS). Currently, drugs administered via this route primarily target the spinal cord and surrounding nerves; however, some drugs can reach the brain to some extent [14]. To the best of our knowledge, there are no reports on the application of intrathecal drug delivery in combination with therapeutic ultrasound. In the present study, we aimed to explore the possibility of delivering genes through this novel drug delivery route, that is, injecting NBs/genes via lumbar puncture and later applying LoFreqUS to the mouse skull. Our goal was to determine whether genes injected into the CSF in the presence of NBs can be “activated” by LoFreqUS irradiation of the skull, resulting in a higher rate of gene transfer.

## 2. Materials and Methods

### 2.1. Preparation of Albumin-Based NBs

A modified procedure based on a previous study was used to generate albumin-based NBs [7]. A schematic of all steps of the preparation of NBs is shown in Figure 1. Initially, 5 mL of perfluoropropane gas (C3F8; sourced from Takachiho Chemical Industry, Tokyo, Japan) was introduced into an empty 5 mL glass vial (Figure 1A). Subsequently, 3.6 mL of degassed distilled water or degassed opti-MEM (Thermo Fisher Scientific, Waltham, MA, USA) solution containing 0.12% human serum albumin (HSA) (Albuminar-25; CSL Behring LLC, Bradley, IL, USA) was poured into the vial (Figure 1A). The vial was securely sealed with a rubber stopper and aluminum cap. An additional 3 mL of C_3_F_8_ gas was injected into the vial for pressurization using a 23-gauge needle inserted through the rubber stopper (Figure 1B). The sealed vials were placed in a high-speed shaking tissue homogenizer (Precellys Evolution; Bertin Instruments, Montigny-le-Bretonneux, France) and shaken at 8300 rpm for 30 s (Figure 1C). This shaking process was performed three times with a 5 min interval to prevent excessive temperature increase. Following the final shaking, the vials were then cooled on ice for 15 min (Figure 1D). An 18-gauge needle was inserted and withdraw once into the rubber stopper of the vial for pressure relief (Figure 1E). The solutions were centrifuged at 120× *g* for 3 min using a MX-301 centrifuge (Takara Tomy, Tokyo, Japan) for removing microbubbles through their flotation and subsequent bursting (Figure 1F). After all the processes were completed, the rubber stopper and aluminum cap were removed, and the NBs solution was collected (Figure 1G,H). A solution containing 0.12% HSA in distilled water or opti-MEM was prepared as a control solution without shaking.

### 2.2. Low-Frequency Ultrasound Irradiation to Albumin-Based NBs

To confirm the response of albumin-based NBs to LoFreqUS, particle size distributions were measured before and after sonication, using a modified version of a previous report [7]. Briefly, NBs (100 µL) were placed in an acoustically transparent film-based 96-multiwell cell culture polystyrene plate (Lumox multiwell; Sarstedt, Nümbrecht, NRW, DE, USA) (Figure 2A). A cell culture plate filled with NBs was positioned above an upward-facing ultrasound transducer with a layer of acoustic coupling gel (Aquasonic 100 gel; Parker Lab., Fairfield, NJ, USA) coated onto an ultrasound gel pad (Sontac; Diagnostic Ultrasound, Bothell, WA, USA) sandwiched between them. The ultrasound irradiation device used had a driving frequency of 47 kHz and an intensity of 1.29 W/cm^2^ measured by a radiation force balance using an ultrasound power meter (UPM-DT-1 and 10; Ohmic Instruments Inc., St. Charles, MO, USA) and a transducer (8 mm diameter). The concentrations of NBs were measured after ultrasound irradiation for various durations (0, 1, 2, 3, and 5 s). Alterations in the number and distribution of NB particles before and after ultrasound irradiation were measured using nanoparticle tracking analysis (NTA) and flow cytometry (FCM).

### 2.3. Characterization of Albumin-Based NBs

The physical properties of the albumin-based NBs were measured using a NTA instrument (NanoSight LM10, Malvern Instruments, Malvern, Worcestershire, UK) and flow cytometric analysis (CytoFLEX; Beckman Coulter, Brea, CA, USA), as previously described [6,7,11,15].

#### 2.3.1. Nanoparticle Tracking Analysis

The motion of the nanoparticles was visualized using light scattering, and Brownian motion was recorded using a sCMOS camera (C11440-50B; Hamamatsu Photonics K.K., Shizuoka, Japan). The camera level was set to level 11 to ensure accurate detection and tracking of the nanoparticles. The shutter and gain of the camera were adjusted to 890 and 146, respectively. The system automatically identified the center of the nanoparticles, tracked the movement of each particle on a two-dimensional plane, and calculated the average distance traveled under Brownian motion, which was recorded for 60 s at 25.0 frames per second. The particle size measurement range for the NTA method was 10–1000 nm. Particle size was estimated using the mean migration distance and the Stokes-Einstein equation. A 0.5 mL solution of NBs diluted with distilled water in the range of 1:30 to 1:50 was injected into the sample measurement chamber of the Nanosight LM 10 system using a 1.0 mL plastic syringe (Terumo, Tokyo, Japan). Nanoparticle suspensions were irradiated with a 638 nm wavelength red laser. Sample image capture and data analysis were performed using application software (NTA 3.4 Build 3.4.4). All experiments for sample measurements were performed independently for each sample. Particle size was reported as mean and mode ± standard error based on triplicate measurements.

#### 2.3.2. Flow Cytometric Analysis

The flow cytometer was equipped with a 405 nm (violet) laser for nanoparticle detection and was configured to measure side scatter (SS) from the violet laser to improve nanoparticle detection. The resolution limit of the violet-SS signal for particle detection was 200 nm. Polystyrene standard beads (200 nm; qNano Calibration Particles; Izon Science, Christchurch, The Netherlands and 500 nm; Archimedes Standard polystyrene beads; Malvern Instruments, Worcestershire, UK) suspended in ultrapure water were pre-measured in a flow cytometer. Violet-SS signals obtained from NBs were analyzed using CytExpert analysis software version 2.4.0.28 (Beckman Coulter, Brea, CA, USA). To determine the size of the NBs produced, gating was based on the size of standard beads in the 200–500 nm range. These data were used to determine the number of NBs in each signal band. The NBs solution was diluted 40-fold prior to measurement.

### 2.4. In Vitro Ultrasound Imaging of Albumin-Based NBs

Ultrasound imaging was performed to confirm the collapse of albumin-based NBs using LoFreqUS. The A-NBs were placed in plastic cuvettes (1 × 1 × 3 cm), and LoFreqUS was applied from the liquid surface for 10 s from the transducer directly immersed in the solution (Figure 3A). The presence of NBs was assessed using diagnostic ultrasound equipment (LOGIQ E9; GE Healthcare Technologies Inc., Chicago, IL, USA) (Figure 3B). To facilitate the ultrasound transmission, a section of the cuvette was replaced with an acoustically transparent PCR film (Diversified Biotech, Dedham, MA, USA). The probe utilized was a Linear Array of 9 MHz with a mechanical index (MI) value of 1.2, operating B-mode with Coded Harmonic Angio (CHA). The solutions underwent stirring using a stirrer, and NBs solutions without sonication, sonicated NBs solutions, and control solutions were all evaluated simultaneously.

### 2.5. Preparation of pDNA and mRNA

The pDNA and mRNA were based on the same construction process as previously reported [6,7].

pNL1.3 CMV-encoding secreted NanoLuc (secNluc pDNA) for in vitro experiments and pGL4.51 [luc2/CMV/Neo] encoding Luc2 (Luc2 pDNA) for in vivo experiments were obtained from Promega (Madison, WI, USA). Amplification of pDNA was performed in Escherichia coli DH5α strain. After isolation, the pDNA was purified using an endotoxin-free plasmid purification kit. The resulting pDNA was dissolved in Milli-Q water and stored at −20 °C until use in each experiment.

DNA templates for in vitro transcription (IVT) of mRNA encoding Gluc (Gluc mRNA) for in vitro experiments and Luc2 (Luc2 mRNA) for in vivo experiments were prepared by inserting protein expression fragments into a pSP73 vector (Promega, Madison, WI, USA) containing the T7 promoter. Prior to insertion, a 120 bp poly A/T sequence was introduced downstream of the protein-coding sequence in the pSP73 vector. This modification allowed us to produce mRNA with a 120 adenine poly(A) tail at the 3’ end by a simple IVT procedure using the pSP73-poly(A) vector. The protein expression fragment was derived from DNA encoding firefly luciferase (pGL4; Promega, Madison, WI, USA).

### 2.6. Cell Culture

The U-87 MG human malignant glioma cell line was procured from the American Type Culture Collection (ATCC) cell bank and cultured in Minimum Essential Medium (MEM; Nacalai Tesque, Kyoto, Japan) supplemented with 10% Fetal Bovine Serum (FBS; Invitrogen, Tokyo, Japan) on a plate coated with 1% gelatin. Cell maintenance was carried out at 37 °C in a humidified atmosphere containing 5% CO_2_. U-87 MG cells were harvested using trypsin–EDTA (Gibco, New York, NY, USA), washed, and immediately placed in fresh medium before each sonoporation experiment. The day before the experiment, the cells were collected and centrifuged at 1000 rpm for 3 min. Subsequently, the cells were seeded at a density of 4.0 × 103 cells per well in an acoustically transparent film based 96-multiwell cell culture polystyrene plate (Lumox multiwell; Sarstedt, Nümbrecht, NRW, DE, USA) coated with 1% gelatin. The cell line was confirmed to be free of viral pathogens, exhibiting an initial viability exceeding 99%, prior to utilization in actual experiments.

### 2.7. In Vitro Cell Sonoporation System

The secNluc pDNA and Gluc mRNA were used for sonoporation of cell lines on multi-well plates, respectively. A schematic of all steps of the experiment is shown in Figure 2. To avoid the influence of ultrasound irradiation on adjacent wells, U87 cells were seeded in a 96-well plate with a minimum spacing of four wells in both vertical and horizontal directions (Figure 2G). Each U-87 MG cells culture medium replaced with 50 μL albumin-based NBs/opti-MEM or control solution, which included 500 ng pDNA or mRNA, respectively (Figure 2B,C). A cell culture plate filled with NBs solution was positioned above an upward-facing ultrasound transducer, with a layer of acoustic coupling gel (Aquasonic 100 gel; Parker Lab., Fairfield, NJ, USA) coated on an ultrasound gel pad (Sontac; Diagnostic Ultrasound, Bothell, WA, USA) sandwiched between them. LoFreqUS was irradiated from transducer to cultured cells (Figure 2D). Conditions of ultrasound were driving frequency 47 kHz, intensity 1.29 W/cm^2^, and ultrasound was irradiated from the transducer (8 mm diameter) The duration of ultrasound irradiation was 0, 1, 2, 3, and 5 s, respectively. After irradiation, the cells were incubated at room temperature for 5 min. The medium sonicated was then removed and replaced with a new incubation medium of the same volume as before irradiation, and the cells were incubated for another 24 to 48 h (Figure 2E,F).

### 2.8. In Vitro Evaluation of Luciferase Expression

In vitro luciferase activity was determined using a Spark Multimode Microplate Reader (Tecan, Männedorf, Zürich, Switzerland). After 48 h of incubation for pDNA and 24 h for mRNA, 10 µL of culture medium was collected from each well of a Costar 96-well white solid plate (Corning, Corning, NY, USA). This was followed by the injection of 1 μg/100 μL of bis-coelenterazine (Gold Biotechnology, Olivette, MO, USA) for secNluc pDNA or coelenterazine (Gold Biotechnology, Olivette, MO, USA) for Gluc mRNA, dissolved in 0.01% Tween 20/0.1 mM EDTA/PBS, into each well. The relative luminescence unit (RLU) values from 2–12 s after injection were plotted and summarized.

### 2.9. Cell Viability Assay

The viability of U-87 MG cells was assessed using a colorimetric method employing 3-(4,5-dimethylthiazol-2-yl)-5-(3-carboxymethoxyphenyl)-2-(4-sulfophenyl)-2H-tetrazolium (MTS). A CellTiter 96 Aqueous One Solution Cell Proliferation Assay System (Promega, Fitchburg, WI, USA) was used. In each well, 20 μL of Cell Titer Solution reagent was added, and a portion of the supernatant was removed for the luciferase assay. After a 2 h incubation period, the absorbance was recorded at 490 nm using a Multiskan Go 96-well plate reader (Thermo Fisher Scientific, Waltham, MA, USA). The survival rate of treated cells was calculated as the ratio of the number of surviving cells in the treatment group to the number of surviving cells in the control group.

### 2.10. Animals

Eight-week-old male BALB/c mice were procured from Japan SLC, Inc. (Hamamatsu, Sizuoka, Japan) and housed in cages within a climate-controlled room. They were maintained on a standard laboratory diet (CE-2; CLEA, Tokyo, Japan) and provided with water. All animal experiments were conducted in compliance with the guidelines for animal experimentation at Fukuoka University, to ensure that ethical standards were met.

### 2.11. In Vivo Intrathecal Administration and Ultrasound Irradiation

Intrathecal administration was performed using a modified version of a previously described technique [16]. Briefly, the hair of the mouse was removed from near the base of the tail to the top of the head to confirm needle insertion and increase ultrasound penetration. In general, mice anesthetized with isoflurane were positioned such that the hips were lifted by a pedestal (Figure 4A).

First, the distribution of intrathecal dyes was examined. Either 50 or 100 μL of opti-MEM solution mixed with 3% Fast Green FCF dye was administered intrathecally between the L5 and L6 vertebral arches of the mice over the course of 1 min using an insulin syringe with 1 mL capacities and 8 mm × 30 G needles (BD Ultra-Fine; Becton, Dickinson and Company, Franklin Lakes, NJ, USA). After administration, the mice were killed by carbon dioxide absorption under general anesthesia. The skull and spinal column were dissected to expose the brain and spinal cord for dye distribution.

Subsequently, experiments were conducted by medullary injection of solutions containing pDNA or mRNA and ultrasound irradiation of the head. In the same manner as the Fast Green CFC dye administration, 60 μL of albumin-based NBs opti-MEM solution or control solution containing 20 μg Luc2 pDNA or 10 μg Luc2 mRNA was administered to the mice. Immediately after administration of the genes, the head of the mouse bent forward 60° was secured by firmly holding the zygomatic arches bilaterally. The transducer was secured at the intersection of the interparietal and occipital bones at the midline of the mouse, with its axial direction aligned towards the tip of the nose. LoFreqUS was repeated thrice for 2 s each time (Figure 4B). After awakening from anesthesia, the mice were allowed to move freely in the cage until the luciferase expression was evaluated.

### 2.12. In Vivo Evaluation of Luciferase Expression

In vivo evaluation of luciferase expression was performed 24, 48, and 120 h after sonication. The mice were anesthetized with isoflurane and placed in In Vivo Imaging System (IVIS Lumina 2) chamber (Caliper Life Sciences, Cheshire, UK) after an intraperitoneal injection of 3 mg/body D-luciferin (Nacalai Tesque, Inc., Kyoto, Japan) as the substrate. Images were acquired using a CCD camera 20 min after the injection of the substrate. Data analysis was performed using Living Image Software 3.2 (Caliper Life Sciences, Cheshire, UK).

### 2.13. Statistical Analysis

Data are presented as mean ± standard error of the mean (s.e.m). Data were analyzed using one-way analysis of variance (ANOVA) with Tukey’s multiple comparisons post hoc test or unpaired *t*-test. Statistically significant differences between groups were analyzed using EZR 1.54 (Saitama Medical Center, Jichi Medical University, Saitama, Japan) [17]. Statistical significance was set at *p* < 0.05.

## 3. Results

### 3.1. NBs Collapse by LoFreqUS Irradiation

The extent of NB collapse induced by ultrasound irradiation was measured using NTA. The NTA results are shown in Figure 5A. Irradiation of the NBs solutions for 1, 3, and 5 s resulted in an alteration in the diameters of NBs to 260.9 ± 14.2 nm, 291.5 ± 16.6 nm, and 288.7 ± 5.5 nm, respectively, from an initial diameter of 272.3 ± 9.7 nm (Figure 4A).

In FCM measurements, sonication for 1, 2, 3, and 5 s reduced the total number of NBs from 23.0 × 10^8^/mL to 19.6 × 10^8^/mL (85.2%), 16.2 × 10^8^/mL (70.3%), 17.9 × 10^8^/mL (77.9%) and 16.6 × 10^8^/mL (72.2%), respectively (Figure 5B, Table A1). The percentage of NBs larger than 500 nm in diameter, NBs 200–500 nm in diameter, and NBs < 200 nm in diameter remaining after 5 s of LoFreqUS irradiation were 22.9%, 67.2% and 95.9% and 95.9%, respectively.

Upon ultrasound imaging of the albumin-based NBs solution, high-intensity echogenic signals were observed which are likely to be bubbles (Figure 6A). In contrast, in NBs solution-irradiated LoFreqUS, the echogenic signals almost disappeared (Figure 6B, Video S1). We were able to demonstrate that the bubbles could be collapsed by ultrasound irradiation, however, the size of the bubble structures was unclear in this experiment.

### 3.2. Gene Transfer by LoFreqUS

The activity of the secNluc protein, which was expressed from pDNA introduced into cells by albumin-based NBs and 1, 2, 3, and 5 s of sonication, was measured as relative luminescence units (RLU). The recorded RLU values were (2.47 ± 0.16) × 10^7^, (1.80 ± 0.60) × 10^7^, (2.50 ± 0.59) × 10^7^, and (2.67 ± 0.16) × 10^7^, respectively (Figure 7A). The RLU values of the group without NBs at 1, 2, 3, and 5 s sonication were (0.08 ± 0.01) × 10^7^, (0.33 ± 0.07) × 10^7^, (0.78 ± 0.37) × 10^7^ and (0.62 ± 0.09) × 10^7^, respectively. The mean values at 1, 2, 3, and 5 s were 31.60 (*p* = 0.00001), 5.50, 3.20, and 4.29 (*p* = 0.00006)-fold higher, respectively, than those of the group without NBs.

Luciferase assay in the microscale in vitro sonoporation of Gluc mRNA with different sonication durations, with and without albumin-based NBs are shown in Figure 7B. The RLU values of the group with NBs at 1, 2, 3, and 5 s were (4.37 ± 1.04) × 10^7^, (4.88 ± 1.35) × 10^7^, (6.04 ± 0.79) × 10^7^, and (6.08 ± 0.71) × 10^7^, respectively, compared to the group without NBs; the mean values at 1, 2, 3, and 5 s were 2.50, 1.8, 2.99 (*p* = 0.0068) and 2.33-fold (*p* = 0.0366), respectively, compared to the group without NBs.

Figure 8 indicates the cell viability assay in cells after pDNA transfection at different sonication times, with and without NBs. The cell viability did not change regardless of the presence or absence of NBs or the duration of sonication. Cell viability at 5 s irradiation time was 91.4 ± 6.12% or 94.4 ± 3.82% with and without NBs, respectively (*p* = 0.4444).

### 3.3. Intrathecal Administration of Dye by Lumbar Puncture

After intrathecal administration of the Fast Green CFC dye via lumbar puncture under general anesthesia, a dissection was performed. Dye injected from the lumbar spine rapidly reached the cranium and accumulated on the surface of the turbinates of the nasal cavity (Figure 9A). The cerebral surface of the basal part of the brain and cerebral arteries were stained (Figure 9B,C). The dye solution reached into the lateral ventricles and into the brain parenchyma at the base of the brain (Figure 9D).

### 3.4. Gene Expression after Intrathecal Administration and Intracranial LoFreqUS Irradiation

Although slight skin impairment was observed, the combination of albumin-based NBs and LoFreqUS yielded sufficient expression of the introduced genes. Figure 9 shows the luciferase expression of cranial region after Luc2 pDNA administration with and without sonication and with and without NBs. Bioluminescence in the occipital and nasal areas irradiated with LoFreqUS was detected only in animals where both administration of NBs and sonication were performed (Figure 10A). Twenty-four hours after sonoporation, robust luciferase expression was detected in the animals in which the intracranial space was irradiated with NBs and LoFreqUS (Figure 10B). Sporadic slight luminescence was detected at the puncture site and along the spinal canal.

The animals with NBs exhibited a 10.05-fold increase in bioluminescence in the cranial region 24 h after sonoporation compared to animals without NBs (*p* = 0.0018). The bioluminescence of the LoFreqUS-irradiated animals without NBs administration was 1.26-fold higher than that of the animals with NBs administration alone (*p* = 0.9930). At 24 and 120 h after sonoporation, luminescence decreased in all groups. Even 120 h after sonoporation, luciferase expression was detected in the group that received the combined treatment with NBs and LofreqUS.

Luciferase expression coded in mRNA after in vivo sonoporation, with and without ultrasound irradiation and with and without albumin-based NBs are shown in Figure 11. Twenty-four hours after sonoporation, luciferase expression was not detected in the cranial area irradiated with LoFreqUS, but slight luciferase expression was sporadically observed at the puncture site and along the spinal canal (Figure 11A). In quantification of bioluminescence, none of the animals detected positive luminescence in the cranial region with/without administration of NBs and with/without LoFreqUS irradiation (Figure 11B).

Percutaneous irradiation of the occipital region of the mice with LoFreqUS resulted in immediate skin damage (Figure A1A). After 24 h, the damage had mostly healed (Figure A1B).

## 4. Discussion

In the past decade, extensive research has been conducted to modify microbubbles and NBs to enhance the delivery efficiency of drugs, which include genes that must be carried to the target lesion to induce genetically altered outcomes for the desired treatment [3]. However, many problems and obstacles must be addressed to achieve this goal. The initial objective of our study was to alter the ultrasound frequency, in this case, to use LoFreqUS that has not yet been studied in depth, and secondly, to minimize the bubble size to a submicron scale, and lastly, to explore an alternative route of gene administration bypassing the tough to cross BBB. The combination of these three factors was proven to be successful to some extent in our study. We demonstrated that, at least in the in vitro experiments, the existence of albumin-based NBs and LoFreqUS irradiation achieved the intracellular delivery of carrier-free genes, avoiding degradation within the experimental protocol timeframe. In the in vivo study, we utilized LoFreqUS to obtain deeper tissue penetration to the brain and also employed a less invasive intrathecal gene/bubble delivery via lumbar puncture, which we dubbed as the “spinal back-door approach”, an alternative to the “front door approach” requiring disruption of the BBB.

To date, many neurological diseases have no curative treatment despite a poor prognosis. These neurological diseases include brain tumors [18], such as glioblastomas, in which genetic mutations in cells lead to abnormal growth and neurodevelopmental disorders resulting from disorders in the formation of neural circuits [19,20]; neurodegenerative diseases such as ALS [21] and Parkinson’s disease [22,23]; Parkinson’s syndrome [24,25]; spinocerebellar degeneration [26,27,28] resulting from the accumulation of pathogenic proteins in neurons and glial cells; and stroke, such as cerebral hemorrhage, due to the blockage or rupture of blood vessels that feed the brain [29].

Gene therapy is expected to be an effective treatment for many neurological diseases [30,31]. In particular, mRNA without the risk of genomic insertion is a safe and innovative therapeutic modality [32]. Unlike pDNA, mRNA has the advantage of functioning in cells with low mitotic potential. Furthermore, mRNA drugs can be designed and manufactured quickly and easily if the gene sequence of the protein of interest is known. These characteristics have led to research efforts to utilize mRNA as a therapeutic modality for CNS diseases [33,34,35,36]. Delivering carrier-free mRNAs with low cell membrane permeability and easy degradation into the cell is difficult, especially when mRNA is administered via the bloodstream and must cross the BBB, which acts as a protective barrier, preventing many substances, including large molecules such as mRNA, from entering the brain. However, developing a safe and effective method for delivering mRNA across this barrier remains challenging. In response to these technical issues, we recently reported a combination of high-frequency ultrasound and NBs as an effective means of carrier-free mRNA delivery into cells [6,7]. NBs have been proven to be very effective in increasing gene transfer to tumor cells. However, there is a need to temporarily disrupt the BBB to deliver genes to the intracranial tissues.

However, the exact mechanism by which NBs enhance drug delivery remains unknown. Several physical experiments, acoustic calculations, and simulation studies have been conducted. Based on previous studies on the Minnaert resonance and its evolving principles, the diameter of a bubble is inversely proportional to the frequency of the ultrasound waves that resonate with the bubble. [37,38]. Based on these principles, it was determined that a very high ultrasound frequency is suitable for resonating NBs. Our previous study ultrasound at a frequency of 1 MHz efficiently collapsed NBs > 200 nm in diameter [6]. However, a recent study has reported that NBs disappear with convergent ultrasound at a low frequency of 80 kHz, which is below the MI 1.9 [39]. Moreover, the combination of NBs and LoFreqUS at 250 kHz with peak negative pressures of 300–500 kPa has been reported to enhance the intracellular delivery of small molecular compounds and effectively reduce the viability of cultured tumor cells [40]. Alternatively, targeted NBs and LoFreqUS have been reported to enhance siRNA delivery into hepatocellular carcinoma cells [41]. In the present NBs collapse experiments, the disappearance of NBs > 200 nm by LoFreqUS with very short durations in the first few seconds was consistent with the results of previous studies. This phenomenon, which does not necessarily conform to the law of ultrasonic responsiveness of bubbles in physical calculations, indicates that NBs may have an unexplainable physical property other than the bubble size above the microscale. Nevertheless, the results of our experiments about characterization of albumin-based NBs and successful in vitro and in vivo gene transfer strongly suggest that LoFreqUS-collapsed submicron-sized bubbles are responsible for the increased gene transfer.

On the other hand, NBs < 200 nm are difficult to collapse by 50 kHz ultrasound irradiation, which is similar to the response of NBs to 1 MHz ultrasound in our previous study [6]. A previous study showed a mathematical model for generation and reduction, in which NBs collapsed by ultrasound irradiation become cavitation nuclei of less than 30 nm again, and grow and collapse repeatedly as NBs [42]. The fact that the concentration of NBs < 200 nm does not decrease regardless of the frequency at which they are irradiated suggests that they have reached equilibrium due to the ultrasound-induced growth and collapse cycle rather than being resistant to ultrasound collapse.

Gene delivery methods and materials are important when implementing gene therapy for many CNS diseases with a poor prognosis. Although carrier-free genes are simple and attractive therapeutic tools, they are difficult to deliver across the cell wall into the cytoplasm. Extracellular DNA and mRNA are foreign molecules rapidly degraded by nucleases [43,44].

It is particularly difficult to deliver drugs and genes to the CNS using an intravascular approach. CNS gene delivery presents a significant challenge primarily attributed to the formidable obstacle known as the BBB. The BBB serves as a crucial physical barrier essential for maintaining CNS chemical homeostasis and safeguarding it against harmful intruders such as bacteria and toxins, as noted in previous studies [45,46]. The BBB is primarily composed of microvascular endothelial cells. This vascular endothelial layer boasts tight junctions between its constituent cells and exhibits limited vesicular transport capabilities, a distinctive feature exclusive to the BBB [45,47,48]. Various chemical carrier or physical energy methods have been attempted as straightforward ways to break through this indestructible barrier head-on [12]. A therapy that uses focused ultrasound and microbubbles to temporarily cleave the BBB and deliver anticancer drugs to tumors in the brain is under clinical investigation as an effective treatment for glioblastoma. From a neurosurgical perspective, utilizing HIFU to facilitate the targeted delivery of chemotherapy drugs and immunotherapeutic agents directly into brain tumors may become a means to avoid the surgical removal of malignant tumors. Researchers are currently exploring ways to harness the precision and targeting capabilities of HIFU to improve outcomes, reduce side effects, and potentially offer new therapeutic options for patients with brain tumors [49].

Alternatively, methods have been attempted to deliver genes to the brain by bypassing the BBB, such as the nose-to-brain pathway, intrathecal administration, intraparenchymal administration, and intracerebroventricular administration [12]. In our study, intracranial gene transfer was achieved from gene administration through the spinal cavity of the lumbar spine by a “spinal back-door approach” technique that bypasses the BBB. Lumbar puncture is a general clinical procedure that is much less invasive than direct brain puncture via trepanation [50]. Several studies have shown that intrathecal administration via lumbar puncture is a promising gene delivery route for treating diseases of the brain, spinal cord, and peripheral nerves [51,52,53,54,55]. The activity of nucleases, especially ribonucleases in normal cerebrospinal fluid is very low [56,57]. This fact may have made our pDNA delivery experiments successful and suggests that the medullary cavity is a promising route for delivering naked genes into the intracranial cavity. In contrast, the composition of RNase in cerebrospinal fluid is different from that in blood [58,59]. These facts suggest that, while the medullary cavity is a promising delivery route for naked genes to the intracranial space, there is a need to consider the complex degradability considerations that differentiate it from serum. This suggests that a different and more complex degradability than that of serum must be considered. In our study, this might have been a barrier to the delivery of degradable mRNAs.

Our results showed that the Fast Green FCF dye administered to the lumbar spinal cord space was delivered intracranially around the nasal cavity and cerebral arteries. Gene transfer efficiency was enhanced in the intracranial and nasal portions of the reach by using a combination of NBs and LoFreqUS. This result can be explained by the results of previous studies on the circulation and excretion kinetics of substances in the spinal fluid. First, the pathways through which the dye and genes administered into the spinal fluid reach the intracranial region can be explained. Intrathecally administered substances passively diffuse into the intrathecal space via convective currents in the spinal fluid [60,61]. The direction of spinal fluid flow in the cerebrospinal cavity is quite simple compared to the flow of blood in blood vessels, and it would easily reach the brain passively without involving the systemic circulation. Drugs and genes administered simply through the lumbar spinal cavity are distributed predominantly in the spinal cord rather than intracranially [62,63]. Previous studies have reported that the amount of intracranial drug solution administered through the spinal cavity of the lumbar spine is proportional to the volume of the solution and the time elapsed since administration [64]. Second, the pathways by which the dye or gene reached the cranium and accumulated at a particular site can be described. It has been conventionally advocated that CSF drains into the blood (venous system) via cranial arachnoid granules (CAGs) and spinal arachnoid granules (SAGs) [65]. Several novel CSF/interstitial fluid excretory pathways have been discovered, including the olfactory route for cerebrospinal fluid drainage into the peripheral lymphatic [66,67], intramural periarterial drainage (IPAD), and glymphatic drainage systems in recent years [68]. Our results suggest that the dye or gene reached the nasal and cranial cavity through these pathways.

The brain, a sophisticated and intricate organ, is housed within the skull and is characterized by its reflective and complex nature. The skull comprises three distinct layers of bone consisting of two cortical plates that sandwich a slender layer of trabecular bone [69]. As ultrasound waves traverse the skull, they undergo reflection, diffraction, and attenuation within the head, leading to a diminished intensity of these waves as they propagate [70,71]. The LoFreqUS can penetrate the brain well, whereas MHz ultrasound cannot penetrate the skull [69]. The drug delivery system (DDS) using LoFreqUS is advantageous for reaching the CNS encased in a rigid spine or skull. In addition, LoFreqUS can be used for BBB opening in combination with microbubbles [9]. With respect to the distribution of transduced genes, NBs and LoFreqUS appeared to enhance the intracellular transduction of genes in a wide range of directly irradiated intracranial tissues or in the noses of indirectly resonant excretory pathways. BBB opening, which uses intravascular bubbles and a relatively high-frequency focused ultrasound, is suitable for gene delivery to deep brain regions, whereas the “spinal back-door approach” using LoEreqUS may be suitable for gene delivery to a wide range of brain surface tissues bordering the medullary cavity. Furthermore, this hypothesis may indicate that intracranial gene delivery using the “spinal backdoor approach” and LoFreqUS is more suitable for treating diseases with widespread lesions, such as neurodevelopmental disorders, which are disorders of neurons in a wide range of the neocortex of the brain surface, and neurodegenerative disorders, such as Alzheimer’s disease, than diseases with local lesions, such as brain tumors and cerebral vascular disorders.

The positive highlight of this study from a material aspect is that no adverse events were observed in any of the animals in which NBs were intrathecally injected via lumbar puncture into the CSF and eventually reached the brain. Only a limited number of materials can be injected into the CSF, some of which result in highly toxic and life-threatening outcomes. Fortunately, the albumin shell or gas within the bubbles did not induce adverse events in the mice. An albumin-based NBs has been administered directly into the CSF for the first time in our study. This nanomaterial has the potential to be applied to a wide variety of imaging modalities, as well as therapeutic ultrasound in the spine or brain.

One of the limitations of the present study animal model is the need for high doses of intrathecal drugs, which are difficult to control before reaching the intracranium. The delivery of mRNA to the intracranial tissues may not have been achieved in the current experiments due to the lack of adequate control of the drug dosage within the safe ultrasound intensity range. In our experiments, more than 50 µL of drug solution was injected in order to reach sufficient intracranial drug concentrations. This amount is well above that reported in existing studies on gene delivery to the spinal cord via lumbar puncture In adult male BALB/c mice weighing approximately 30 g [72]. If this dose were extended to a human weighing 60 kg, more than 100 mL of the drug solution would be required. This amount is too large to be applied in a realistic world. Nevertheless, gene transfer of pDNA using the same experimental method of intrathecal gene/NBs delivery and ultrasound irradiation under identical acoustic settings resulted in marked intracranial gene transfer. Despite our excellent pDNA results, the same was not observed in the mRNA experiments. It was again shown that the combination of NBs and LoFreqUS is indeed an effective intracellular delivery method for naked mRNA in vitro. On the other hand, the slight luminescence observed at the puncture site and along the spinal canal in the gene transfections experiments in vivo was not associated with the combination of NBs and LoFreqUS and appeared to have been the result of hydrodynamic transfection by injection of high doses of fluid into the intrathecal space [73,74]. Again, the results are significant, demonstrating that while mRNA can be slightly delivered to the spinal cord, its intracranial delivery does not mirror that of DNA. There is a need to address this issue, such as adding modifications to target the delivered gene or changing the bubble formulation to affiliate with the brain tissue, which may lead to a reduction in the total amount of drug administered. These technical modifications may result in a more successful mRNA transfer in future studies.

An additional limitation of the present study was the continuous LoFreqUS irradiation, which caused minor skin damage. Previous reports have shown that 0.5 s of ultrasound irradiation at 20–60 kHz achieved safe delivery of naked mRNA to the intestinal mucosa of mice [75]. As the skin damage was reversible and the mice eventually recovered, future optimization of the acoustic conditions of LoFreqUS irradiation would further improve the gene delivery efficiency and safety of the subject.

## 5. Conclusions

The present study is the first to report successful intracranial gene transfection by intrathecal administration of nanobubbles/gene (“spinal back-door approach”), followed by irradiation to the skull with low-frequency ultrasound. Although in vivo experiments showed positive results only for pDNA and not for mRNA, we believe that this is the first major step forward that could become an innovative therapeutic tool against CNS diseases with poor prognosis.

## Figures and Tables

**Figure 1 nanomaterials-14-00285-f001:**
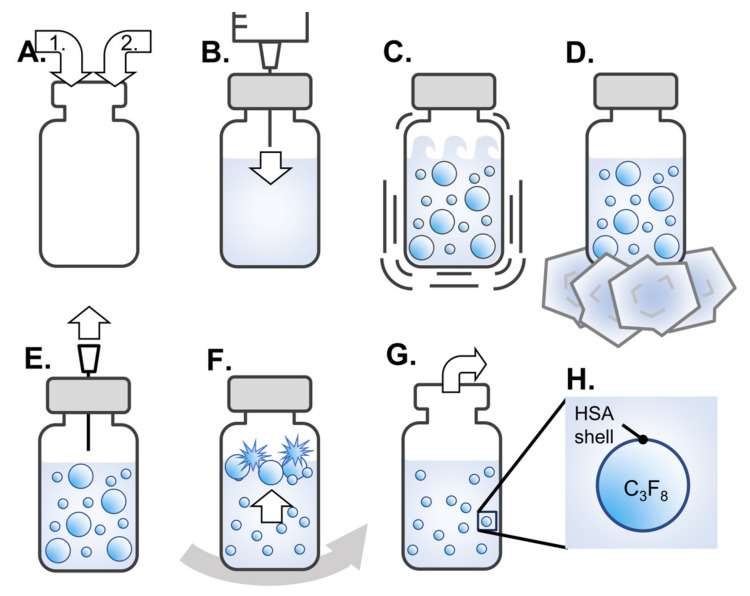
Schematic representation of preparation of albumin-based NBs solutions. (**A**) Gas replacement with perfluoropropane gas (white arrow 1) and injection of distilled water or opti-MEM containing 0.12% HSA (white arrow 2) into vial. (**B**) Pressurization by addition of perfluoropropane gas into a vial sealed with a rubber stopper and aluminum cap (white arrow). (**C**) Bubbling by high-speed shaking. (**D**) Cooling for bubble stabilization. (**E**) Pressure relief inside the vial (white arrow). (**F**) Centrifugation (gray arrow) to float and burst microbubbles (white arrow). (**G**) Recovery of solution containing NBs from opened vials. (**H**) Schematic of the structure of albumin-based NBs. NBs: nanobubbles. HSA: human serum albumin. C_3_F_8_: perfluoropropane gas.

**Figure 2 nanomaterials-14-00285-f002:**
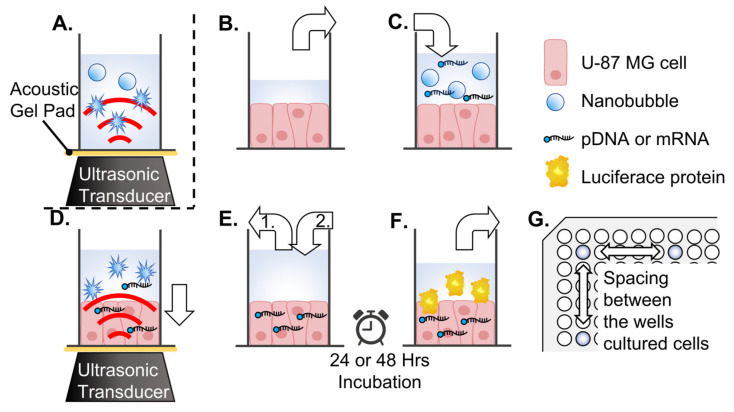
Schematic representation of ultrasound treatment of albumin-based NBs or sonoporation in 96-well plates. (**A**) Sonication treatment of NBs by applying ultrasound to the medium. (**B**–**F**) Sonoporation method. (**B**) Incubation medium is removed from the well (white arrow) of a 96-well plate seeded with malignant glioma cells (U-87 MG). (**C**) Each well is filled with medium containing NBs (white arrow). (**D**) Transfection by ultrasound. pDNA or mRNA is transferred into the cytoplasm (white arrow). (**E**) Aspiration of sonicated medium (white arrow 1) and (**E**) addition of fresh incubation medium (white arrow 2). (**F**) After 24 or 48 h of incubation, the supernatant was collected for reporter assay. (**G**) Arrangement of wells with seeded cells (indicated by color) on a 96-well plate. NBs: nanobubbles.

**Figure 3 nanomaterials-14-00285-f003:**
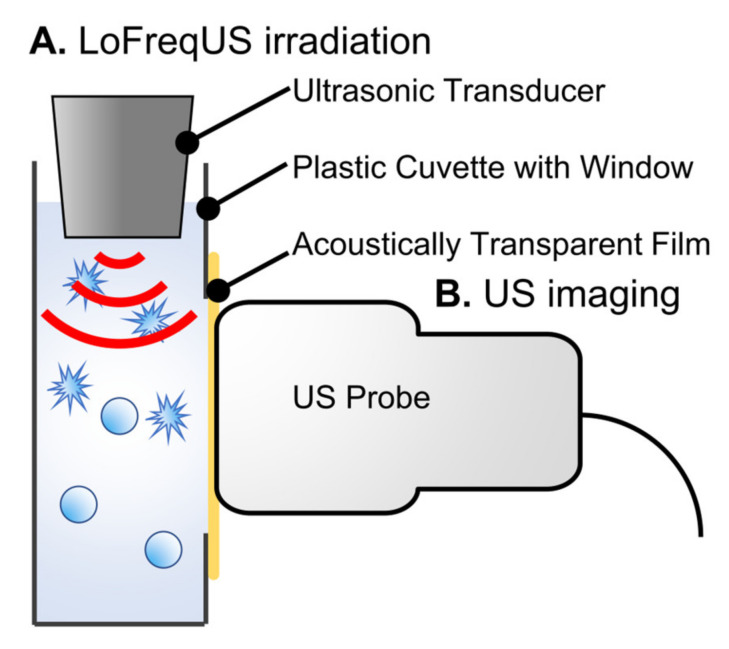
Overview of in vitro ultrasound imaging of albumin-based NBs. (**A**) LoFreqUS irradiation into A-NBs solution. (**B**) Acoustic evaluation using a diagnostic US imaging system. NBs: nanobubbles, US: ultrasound.

**Figure 4 nanomaterials-14-00285-f004:**
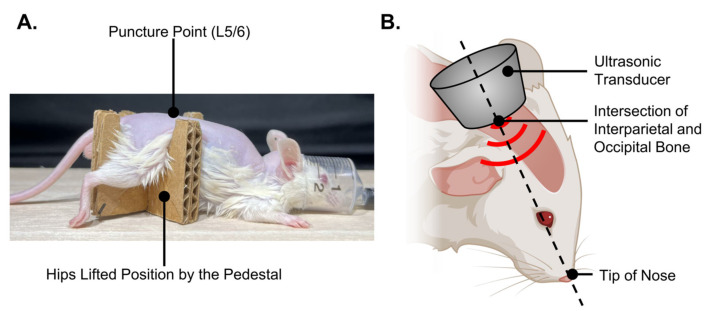
Overview of intrathecal lumbar puncture and in vivo sonoporation. (**A**) Posture of a mouse in which a lumbar puncture was performed. (**B**) Schema of sonication method to the cranial region of the mouse.

**Figure 5 nanomaterials-14-00285-f005:**
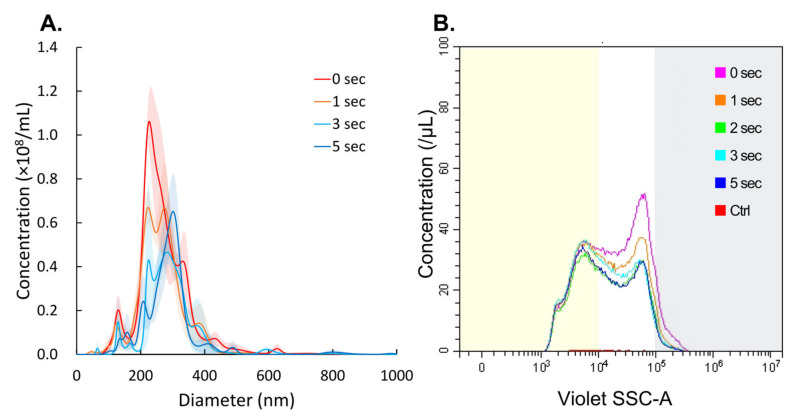
Physical properties of NBs at different ultrasound exposure times. (**A**) Size distribution of NBs on NTA. (**B**) FCM measurement NBs size distribution and concentration after 40-fold dilution. Yellow or gray shaded area indicates NBs size less than 200 nm or more than 500 nm, respectively. NBs: nanobubbles.

**Figure 6 nanomaterials-14-00285-f006:**
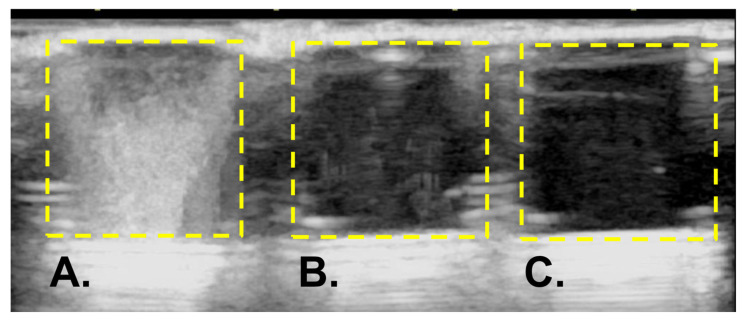
Visualization of albumin-based NBs using ultrasound diagnostic equipment. (**A**) Image of NBs solution before LoFreqUS irradiation. (**B**) Image of NBs solution after LoFreqUS irradiation. (**C**) Image of control solution. NBs: nanobubbles.

**Figure 7 nanomaterials-14-00285-f007:**
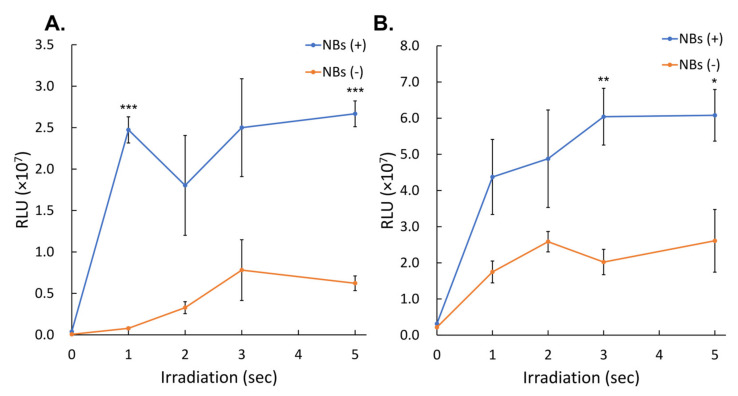
pDNA or mRNA transfection efficiency by sonoporation using different irradiation times. (**A**) Ultrasonic irradiation time-dependent profile of pDNA luciferase expression in the condition with or without albumin-based NBs. (**B**) Ultrasonic irradiation time-dependent profile of mRNA luciferase expression in condition with or without NBs. Data are presented as mean ± standard error of the mean (s.e.m.). Statistical significance was assessed using unpaired *t*-test (* *p* < 0.05, ** *p* < 0.01, *** *p* < 0.001) (*n* = 4). NBs: nanobubbles, RLU: relative luminescence units.

**Figure 8 nanomaterials-14-00285-f008:**
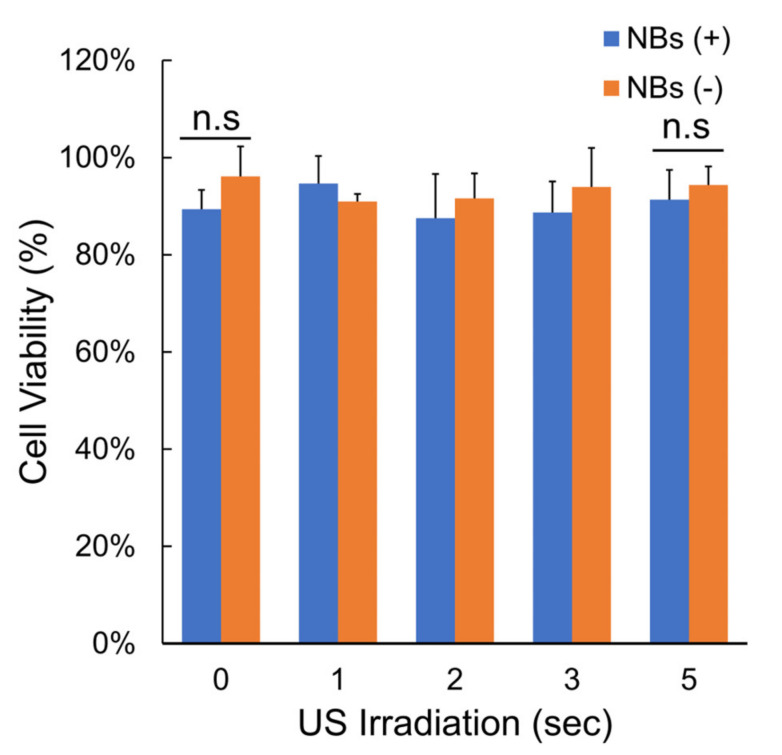
Cell viability after sonication. Cell viability with and without NBs and after different ultrasound exposure times, data are presented as mean ± standard error of the mean (s.e.m.). Statistical significance was assessed by unpaired *t*-test (n.s. not significant) (*n* = 4). NBs: nanobubbles.

**Figure 9 nanomaterials-14-00285-f009:**
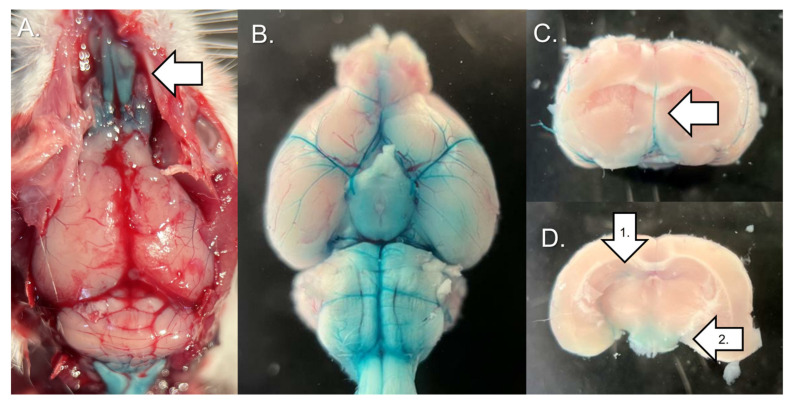
Distribution of intrathecal dye after administration via lumbar puncture. (**A**) Overall view of the cranial and nasal cavity with the skull removed (100 μL dye injected; arrow indicates the turbinates). (**B**) Whole brain viewed from below. (**C**) Coronal section of the cerebrum in the region anterior to the hypothalamus (50 μL dye injected; arrow indicates the anterior cerebral artery). (**D**) Coronal section of the cerebrum in the center of the hypothalamus (50 μL dye injected; arrow 1 indicates the lateral ventricle, arrow 2 indicates the hypothalamus).

**Figure 10 nanomaterials-14-00285-f010:**
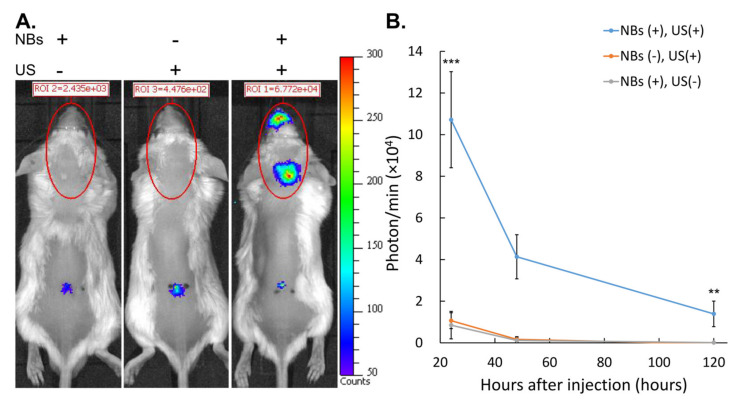
Intracranial luciferase expression in mice after intrathecal injection of pDNA. (**A**) Representative IVIS images of mice one day after sonoporation. (**B**) Quantification of luciferase expression of the cranial region over time based on IVIS images. Data are presented as mean ± standard error of the mean (s.e.m.). Statistical significance was determined using a one-way ANOVA (** *p* < 0.01, *** *p* < 0.001), NBs+US− (*n* = 4), NBs−US+ (*n* = 6), NBs+US+ (*n* = 5). NBs: nanobubbles, US: ultrasound.

**Figure 11 nanomaterials-14-00285-f011:**
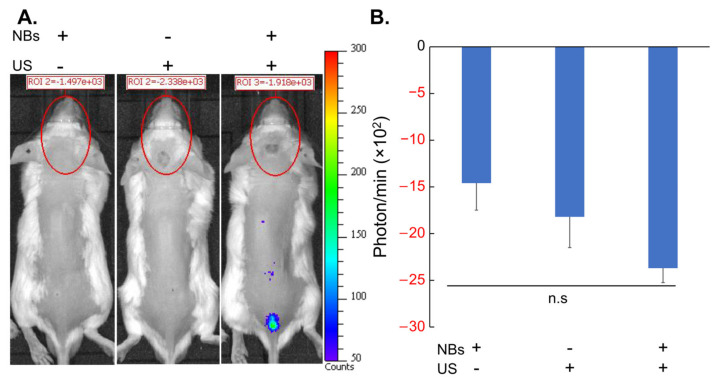
Intracranial Luciferase expression in mice after intrathecal injection of mRNA. (**A**) Representative IVIS images of mice one day after sonoporation. (**B**) Quantification of luciferase expression of the cranial region on IVIS images. Data are presented as mean ± standard error of the mean (s.e.m.). Statistical significance was determined using a one-way ANOVA (n.s. not significant), NBs+US− (*n* = 3), NBs−US+ (*n* = 4), NBs+US+ (*n* = 4). NBs: nanobubbles, US: ultrasound.

## Data Availability

Data are contained within the article and Appendix A.

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
