# Peer review of "Intracranial Gene Delivery Mediated by Albumin-Based Nanobubbles and Low-Frequency Ultrasound"

_nanomaterials, 2024, doi:10.3390/nano14030285_

Round 1
Reviewer 1 Report
Comments and Suggestions for Authors
The author introduced a novel direct intracranial gene delivery method that bypasses BBB using human serum albumin-based nano-bubbles injected via a less invasive intracranial route via lumbar puncture and then performs intracranial irradiation with low-frequency ultrasound. This article is well-written and well-supported by the data presented. Although the provided results appear to be conclusive, some comments, as listed below, must be addressed.
1. NTA data does not show 'dilution magnification' (Figure 4)
If the concentration is too high, it seems to have been diluted because it cannot be measured. Also, NTA's condition should be attached (camera level, shutter time, camera gain value, etc.)
2. Table A1 shows that 200nm bubbles do not respond to ultrasound. Are only bubbles larger than 500 nm sensitive to ultrasound? Need more explanation.
Comments on the Quality of English LanguagePlease check some typos.
Author Response
Thank you very much for your insightful suggestion and positive comments.
1. As you rightly pointed out, information on the conditions used for Nanoparticle Tracking Analysis (NTA) is indeed crucial for accurate assessment but was not included in the original manuscript. These details have now been added to lines 155–178 in the section 2.3.1. Nanoparticle tracking analysis of the manuscript. Furthermore, the description about the type of camera (C11440-50B) used in our study has been corrected from CCD to sCMOS in line 164. To succinctly outline, the conditions for NTA used in this study are as follows:
Hardware: Nanosight LM 10 (Malvern Panalytical, UK)
NTA Version: NTA 3.4 Build 3.4.4
Camera Type: sCMOS
Laser Type: Red (638nm)
Camera Level: 11
Slider Shutter: 890
Slider Gain: 146
FPS: 25.0
Number of Frames: 1498 (≈60 sec at 25.0 frames/sec)
Dilution factor: 1:30~1:50 with distilled water.
2. Indeed, this aspect warrants further exploration. Our previous research (Kida H., et al. Front Pharmacol. 2022 Jun 1;13:855495.) demonstrated that 1MHz ultrasound efficiently collapses nano-bubbles larger than 200nm while maintaining the concentration of those smaller than 200nm. Two hypotheses were considered in that study:
1) Based on the law of resonance between ultrasound frequency and NBs diameter, 1MHz ultrasound might not collapse NBs smaller than 200nm.
2) The nano-bubbles collapsed by ultrasound might serve as cavitation nuclei, potentially regenerating nano-bubbles smaller than 200nm. This could ostensibly result in the maintained concentration of NBs smaller than 200nm, as suggested by Yasuda et al. in their study (Yasuda K., et al. Chemical Engineering Science, 2019, p 455-461.).
The confirmation of a similar phenomenon with ultrasound irradiation below 50kHz in this study is intriguing. For such a low frequency as 50kHz, both nano-bubbles larger and smaller than 200nm are substantially off the calculated resonant diameter. Hence, the results do not necessarily indicate that the collapse threshold diameter of nano-bubbles is 200 nm. The phenomenon of maintained concentration of nano-bubbles smaller than 200nm under both 1MHz and 50kHz US irradiation might suggest the latter hypothesis, where collapsed nanobubbles serve as cavitation nuclei, regenerating new nanobubbles. These explanations have been added to lines 520–527 of the Discussion section.
Other modifications include the following.
[Correction to the description of in vitro sonoporation methods]
Regarding materials and methods, we apologize for the missing content in 2.7, In vitro cell sonoporation system, which was duplicated in 2.8, In vitro evaluation of luciferase expression. We have again described the detailed method of in vitro cell sonoporation in line 242-267.
[Unification of abbreviations]
In previous versions of the manuscript, terms related to nanobubbles were mixed; the full spelling "nanobubbles," the abbreviations "NBs: Nanobubbes," and "A-NBs: Albumin-based nanobubbles".The mixing of these terms may confuse the reader, so all references to "NBs" have been unified except where it is necessary to use the full spelling "nanobubbles". Also, only where the composition of the bubble shells is specifically mentioned, "albumin-based NBs" are described.
[Adittion the correct URL for Supplementary Materials].
We apologize that the URL to Video S1 was not listed in the previous manuscripts. The correct URL is now listed in line 662.
[Uniformity of capitalization and lowercase letters used in titles and headings]
The initial letters of the words in all paragraph headings in Materials and Methods, and Results have been capitalized.
Reviewer 2 Report
Comments and Suggestions for Authors
In this work, the authors presented an intracranial gene delivery approach which leverages low-frequency ultrasound to improve tissue penetration and gene delivery efficiency. The concept and design are attractive to the related research community. However, a few more questions need to be addressed before acceptance of publication.
1. How about the construction of nanobubbles? The authors should draw a scheme illustrating the preparation process and chemical components of nanobubbles.
2. I wonder where the luminescence on the back or tail of mice is coming from in Figure 9A and Figure 10A.
3. In figure 10A, luminescence signal could be hardly found in cranial area.
Author Response
Thank you very much for your insightful suggestion and positive comments.
1. As you suggested, illustrating the specific preparation method and structure of nanobubbles would indeed facilitate the readers' understanding. Following your recommendation, the preparation method and chemical components of the nanobubbles have been added as Figure 1 and its legend (in lines 118-126). Unclear nanobubble preparation procedures were clearly rewritten in line 93-117.It has also been noted in lines 59-60 that these preparation methods and the structure of the albumin shell have been established in our previous research.
2. As you noted, the presence of luminescence on the back or tail of mice, albeit minimal, is significant. These observations at the site of lumbar puncture, the base of the tail where the spinal canal expands, or the thoracic spine were sporadic and not quantitatively analyzable. This minor luminescence was detected irrespective of ultrasound irradiation or the presence of nano-bubbles. These results have been clearly described to lines 422-423(pDNA), 441–442(mRNA) of the manuscript. In the discussion in lines 636-640, It was described that nonspecific luminescence might be a hydrodynamic gene introduction due to the volume administered in this study.
3. Apologies for any confusion regarding the results of mRNA in vivo delivery. Concisely, the results indicate that mRNA was not delivered to the cranial area in this experiment. Figure 10A and B document the absence of luminescence activity due to luciferase in the ROI of the head, signifying a ‘non-detect.’ This negative result is crucial as it underscores the challenges of delivering mRNA to the central nervous system in vivo, as opposed to pDNA. However, we admit that the initial presentation of results was unclear, and the negative values displayed in Figure 10B were confusing. The manuscript has been revised for clearer articulation of the results in line 440-441, and the graph has been replaced to better represent the data with obvious negative values (in Figure 10B). The importance of this result suggesting a difference between transfection of naked DNA and mRNA in vivo was highlighted and described of the discussion, contrasted with the combination of nanobubbles and LoFreqUS being an effective intracellular delivery method for naked mRNA in vitro(lines 634-642).
Other modifications include the following.
[Correction to the description of in vitro sonoporation methods]
Regarding materials and methods, we apologize for the missing content in 2.7, In vitro cell sonoporation system, which was duplicated in 2.8, In vitro evaluation of luciferase expression. We have again described the detailed method of in vitro cell sonoporation in line 239-264.
[Unification of abbreviations]
In previous versions of the manuscript, terms related to nanobubbles were mixed; the full spelling "nanobubbles," the abbreviations "NBs: Nanobubbes" and "A-NBs: Albumin-based nanobubbles".The mixing of these terms may confuse the reader, so all references to "NBs" have been unified except where it is necessary to use the full spelling "nanobubbles". Also, only where the composition of the bubble shells is specifically mentioned, "albumin-based NBs" are described.
[Adittion the correct URL for Supplementary Materials].
We apologize that the URL to Video S1 was not listed in the previous manuscripts. The correct URL is now listed in line 662.
[Uniformity of capitalization and lowercase letters used in titles and headings]
The initial letters of the words in all paragraph headings in Materials and Methods, and Results have been capitalized.
Round 2
Reviewer 2 Report
Comments and Suggestions for Authors
The authors answered my questions and I have no further questions.
Author Response
Thank you very much for your positive comments. Your pertinent suggestions have made our research even more meaningful.